# Remote testing of vitamin D levels across the UK MS population—A case control study

Nicola Vickaryous[1], Mark Jitlal[1☯], Benjamin Meir Jacobs[1☯], Rod Middleton[2], Siddharthan Chandran[3], Niall John James MacDougall[4,5], Gavin Giovannoni[1,6,7], Ruth Dobson[1,7] *

1 Preventive Neurology Unit, Wolfson Institute of Preventive Medicine, Queen Mary University London, London, United Kingdom, 2 UKMS Register, Swansea University Medical School, Swansea, United Kingdom, 3 Centre for Clinical Brain Sciences, UK Dementia Research Institute at Edinburgh, University of Edinburgh, Edinburgh, United Kingdom, 4 Neurology Department, Hairmyres Hospital, East Kilbride, United Kingdom, 5 Neurology Department, Institute of Neurological Sciences, Glasgow, United Kingdom, 6 Blizard Institute, Queen Mary University London, London, United Kingdom, 7 Department of Neurology, Royal London Hospital, BartsHealth NHS Trust, London, United Kingdom

☯ These authors contributed equally to this work.
* ruth.dobson@qmul.ac.uk

**Data Availability Statement:** Individual level data used in this study is available via the UK MS Register by application from any suitably qualified investigator to the UK MS Register steering

## Abstract

### Objective

The association between vitamin D deficiency and multiple sclerosis (MS) is well described. We set out to use remote sampling to ascertain vitamin D status and vitamin D supplementation in a cross-sectional study of people with MS across the UK.

### Methods

People with MS and matched controls were recruited from across the UK. 1768 people with MS enrolled in the study; remote sampling kits were distributed to a subgroup. Dried blood spots (DBS) were used to assess serum 25(OH)D in people with MS and controls.

### Results

1768 MS participants completed the questionnaire; 388 MS participants and 309 controls provided biological samples. Serum 25(OH)D was higher in MS than controls (median 71nmol/L vs 49nmol/L). A higher proportion of MS participants than controls supplemented (72% vs 26%, p<0.001); people with MS supplemented at higher vD doses than controls (median 1600 vs 600 IU/day, p<0.001). People with MS who did not supplement had lower serum 25(OH)D levels than non-supplementing controls (median 38 nmol/L vs 44 nmol/L). Participants engaged well with remote sampling.

### Conclusions

The UK MS population have higher serum 25(OH)D than controls, mainly as a result of vitamin D supplementation. Remote sampling is a feasible way of carrying out large studies.

committee. Further details are available via the UK MS Register site: https://ukmsregister.org/Research/WorkingWithUs, where full contact details and the process for application is described in detail. Ethical constraints set out in the approvals for the UK MS Register limit the sharing of this personal level data without application, due to the individual level sensitive data that is collected and available.

**Funding:** This study was funded by the UK MS Society. The work was performed on the Preventive Neurology Unit, which is funded by Barts Charity. The funders had no role in study design, data collection and analysis, decision to publish or preparation of the manuscript.

**Competing interests:** None of the authors have any financial disclosures or competing interests relevant to this work.

## Introduction

MS susceptibility is a complex trait influenced by genetic and environmental factors. Established environmental risk factors include EBV seropositivity, smoking, and childhood obesity [1–3]. Low serum 25-hydroxyvitamin D (25(OH)D) levels in adulthood, or even soon after birth, are associated with greater risk of developing multiple sclerosis (MS) [4–6]. Vitamin D is primarily derived from the UV light-dependent conversion of 7-dehydrocholesterol to cholecalciferol in skin. Serum 25(OH)D is formed by the hepatic 25-hydroxylation of cholecalciferol, which is further hydroxylated in the kidney to generate the biologically active compound (1,25 hydroxyvitamin D). 25(OH)D is most commonly used as a measure of vitamin D status due to its long half-life, relative stability and direct biological relationship to 1,25 hydroxyvitamin D [7].

Vitamin D is an attractive target for potential intervention in MS as it represents an easily modifiable factor. However, data is conflicting regarding the role of vitamin D in driving inflammation and/or progression in people with established MS. Clinical trials of vitamin D supplementation in MS have failed to provide robust evidence of benefit [8–11]. Several recent meta-analyses looking at clinical trials of vitamin D for the treatment of MS have demonstrated at best modest reductions in annualised relapse rates (ARR) and/or brain lesion activity but no impact on disability [12–14].

There are thought to be multiple factors influencing vitamin D status in MS populations [15]. Current population guidelines recommend an intake of at least 400IU/day vitamin D for all [16]. There is a lack of consensus and evidence on whether people with MS should be advised to supplement with vitamin D over and above the advice given to the general population. Single centre studies examining vitamin D supplementation behaviours are subject to bias due to practices of individual neurologists; collecting supplementing information without the wider lifestyle context or serum vitamin D levels significantly limits interpretation.

Remote sampling using dried blood spots provides a means of testing biomarkers across an entire population without the need for in-person visits, which is of rapidly increasing relevance in the current COVID-19 pandemic. We set out to examine the feasibility of a large-scale research project performed entirely remotely, including remote sampling using dried blood spots. We used remotely deployed questionnaires backed up with biological sampling to examine the behaviours and lifestyle factors that influence vitamin D and assess their contribution to the serum vitamin D status across the UK MS population.

## Methods

### Study recruitment

The primary method of recruitment was via the UK MS Register [17]. 14,991 individuals with MS were invited to participate; 1722 people with MS provided informed consent and completed a baseline questionnaire over 6 weeks using the online platform; an additional 25 participants (postal participants) directly contacted the study site (Fig 1). Individuals were additionally recruited via regional MS networks. Questionnaires with sampling kits were distributed to three MS clinics across the UK—Edinburgh, Lanarkshire and London. 68 sampling kits were handed out to potential participants.

Each MS participant who was given a sampling pack was asked to recruit an unrelated friend as a matched control. They were asked to select someone of the same gender, within 5 years of age and living within a 50-mile radius (but not in the same house) as themselves.

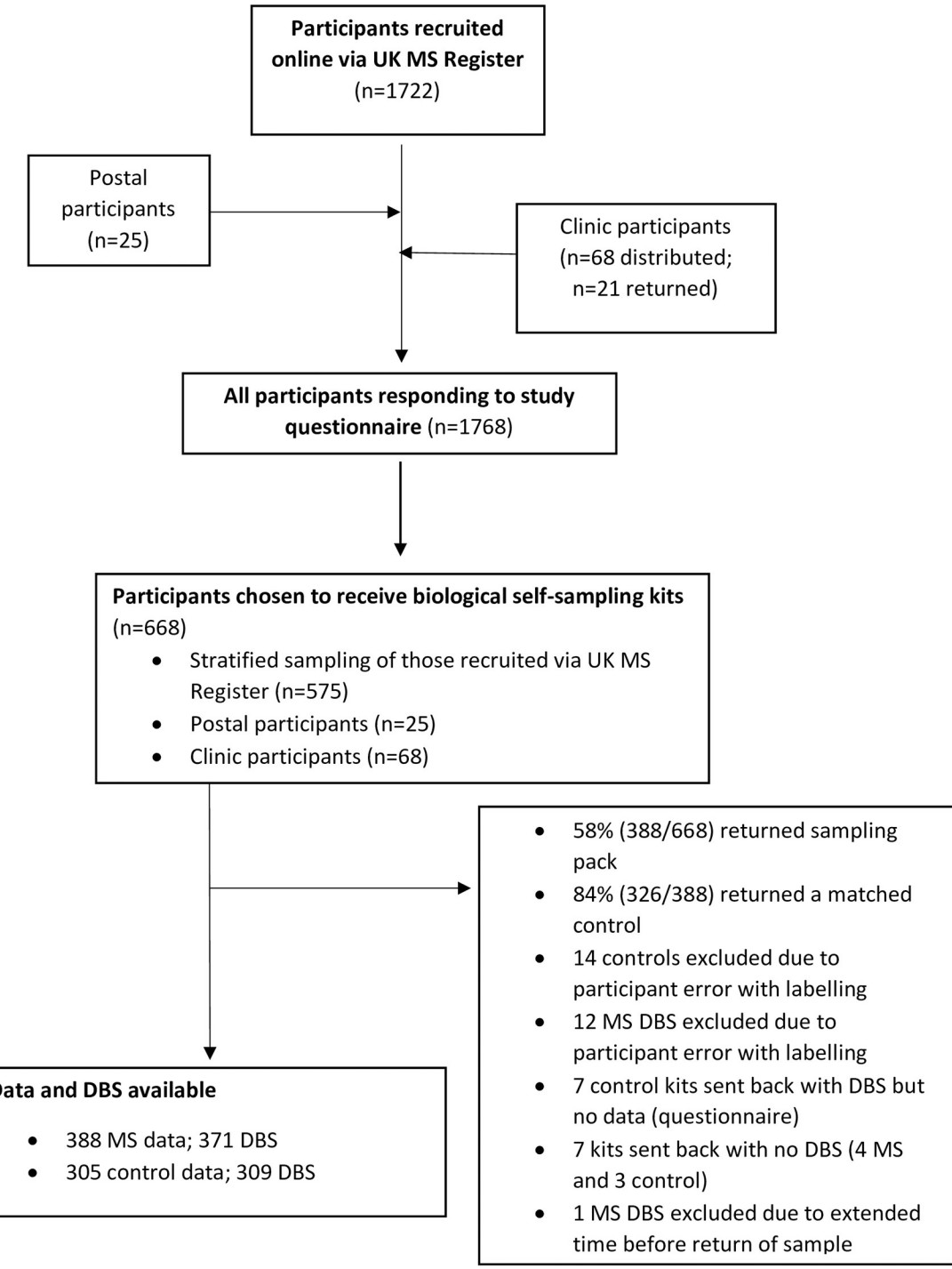

**Fig 1. Flow chart of the study recruitment and resulting study population.** 388 MS participants and 309 matched controls returned sampling kits. In some cases, data or biological material was not available for all data points, resulting in 388 MS with data and 371 MS with DBS and 305 controls with data and 309 controls with DBS.

## Ethical permissions

The UK MS Register has ethical approval via South West Bristol REC (16/SW/0194). This study had additional ethical permissions via London Stanmore REC (18/LO/1455).

## Stratified random sampling

Stratified random sampling was used to select 575 UK MS register participants to receive kits. Participants were grouped (stratified) based on geographical location (100km x 100km square), MS type (RRMS, SPMS, PPMS) and disability (low disability classified as EDSS <6, high disability EDSS ≥6). Random sampling within groups was then performed. This approach attempted to ensure that we had a balance across geographical areas, MS disease types, and EDSS in order that we could make meaningful comparisons between pre-specified subgroups or potential influences of interest. If we had sought to represent the initial respondents we would not have had enough individuals in some cells in order to be able to perform such analyses.

## Questionnaire data

A host of demographic and MS-specific data were collected including geographical location, gender, age, BMI, smoking status, MS type, EDSS, MSIS and date of diagnosis. Where available, Expanded Disability Status Scores (EDSS) derived from a web-based application were used as a proxy for disability levels [18], and estimates of disease physical and psychological impact via the Multiple Sclerosis Impact Scale (MSIS-29). Data on current vitamin D supplementation was collected including supplement use, frequency, and dose at the time of questionnaire completion. Participants completing the online form were invited to upload an image of their supplement to validate supplement dose. Information of diet type and consumption of oily fish, and assessment of time spent on outdoor activities and UV sun protection averaged over the past 3 months was also collected. To ensure complete capture of sun protection factor containing products in addition to sunblock (moisturiser, foundation, mineral powder etc.), participants were asked about both 'cosmetic sunblock' and 'sunblock' usage.

## Sampling kits

Each sampling pack contained two sampling kits, one for the MS participant and one for their matched control. Each sampling kit contained a fully equipped dried blood spot (DBS) sampling system to collect a blood sample for vitamin D analysis, and a buccal swab for genetic material. A questionnaire was included for controls, and for those MS participants where data was not entered via the online system. Sampling packs were sent out February-July 2019. Samples were received back at the study site February-September 2019.

## 25(OH)Vitamin D analysis

Serum vitamin D concentrations were measured from DBS [19]. Upon receipt samples were stored at -80˚C and underwent analysis in four batches. Liquid chromatography tandem mass spectrometry was used to determine total 25(OH)D [20, 21]. Two DBS were analysed per participant; results were excluded if duplicate analysis differed by ≥15%, if only one viable DBS was available, or if DBS were deemed to be of poor quality, i.e, spots too small, not fully soaked through or multiple overlapping spots.

## Vitamin D levels and MS in UK Biobank

We then set out to verify our findings using an independent sample set derived from UK Biobank (UKBB) [22]. Questionnaire and biomarker data from participants' baseline visit (2006–2010) were used. Each individual with MS at the time of UK Biobank registration (n = 1978) was randomly matched to four controls (n = 7912), stratified by age, gender, and ethnicity (white vs non-white). Data including baseline serum 25(OH)D levels, vitamin D supplementation (yes/no; no dose information available), oily fish consumption, time spent outdoors and UV sun protection usage were analysed.

## Statistical analysis

Statistical analyses for MS register data were performed using SPSS v26 and R (v.1.2.5001). Geographical mapping was performed using ArcGIS 10.5. Analysis of UK Biobank data was carried out using R (version 3.6.1). Relationships between categorical variables were analysed using the chi-squared test of association; non-normally distributed continuous variables were analysed using the Mann-Whitney U test and the Kruskal Wallis test was used to compare 3 + groups. Simple linear regression was used to examine the relationship between demographic, solar and lifestyle behaviour that may affect dose of vitamin D and serum 25(OH)D levels.

# Results

## Questionnaire data

1768 participants with MS provided questionnaire data. This group consisted of 1722 individuals recruited via the UK MS Register, 25 postal participants and 21 participants from local MS clinics who returned packs. This group had a wide geographical distribution across the UK (S1a Fig). Their demographics were consistent with that expected across an MS population; 75% female and predominantly relapsing remitting MS (RRMS) (Table 1).

**Table 1. Participant demographics.**

| | All participants[a] | Biological sampling group[b] | | |
|---|---|---|---|---|
| | MS (n = 1768) | MS (n = 388) | Control (n = 305) | p-value |
| female, n (%) | 1329 (75) | 292 (75) | 229 (75) | 1 |
| male, n (%) | 439 (25) | 96 (25) | 76 (25) | |
| age, median (IQR) | 53 (15) | 56 (14) | 55 (16) | 0.37 |
| BMI, median kg/m$^2$ (IQR) | 25; 6 | 25; 6 | 26; 6 | 0.02 |
| current smokers, n (%) | 70; 5 | 16; 5 | 19; 6 | 0.56 |
| **MS type, n;%** | | | | |
| RRMS | 976; 55 | 137; 35 | | |
| SPMS | 459; 26 | 120; 31 | | |
| PPMS | 203; 12 | 93; 24 | | |
| Other | 130; 7 | 38; 10 | | |
| **EDSS** | | | | |
| median; IQR (n) | 6.0; 4 | 6.5; 3 | | |
| low EDSS (<6): n; % | 247; 49 | 41; 38 | | |
| high EDSS (≥6); n; % | 259; 51 | 66; 62 | | |

[a]data was missing for the following; BMI 1374 participants, current smoking status 310 participants, EDSS 1262 participants.
[b]data was missing for the following: Age, 5 MS and 7 control; BMI, 271 MS and 7 control; current smoking status 69 MS; EDSS 281 MS.

## Biological sampling and matched controls

600 sampling kits were posted out to participants. Of 100 kits sent to network sites, 68 were distributed to potential participants. Sampling packs were sent out to participants from across the United Kingdom including the Shetland Islands, Orkney Islands, Outer Hebrides, Isle of Man and Channel Islands (S1b Fig). 388 sample kits (58%) were completed and returned; those who did not return kits were slightly younger and more likely to be male (S1 Table). 326/ 388 returned kits (84%) included a matched control. 17 MS and 17 control participants had DBS samples excluded or not received, and 7 controls did not complete a questionnaire (4 of whom provided a DBS). Thus 388 MS cases (371 with DBS), 309 control DBS, and 305 control questionnaires were included in the analysis (Fig 1).

The demographics of the group from whom biological samples were obtained reflected stratified sampling across MS type and disability levels (Table 1), with approximately 35% RRMS, 31% SPMS, 24% PPMS. EDSS scores were available for 107 participants in this group; the median EDSS was 6.5 (IQR 3) (Table 1). Controls appeared well-matched (Table 1), with no significant difference in sex or age distribution. Controls had a slightly higher BMI than participants with MS (median BMI 25 in MS vs 26 in controls; p = 0.02), and there was no difference in the proportion of current smokers in the two groups.

## Vitamin D supplementation between MS and controls

72% (276/386) of the MS participants from the biological sampling group reported taking vitamin D supplements compared to 26% (79/305) of controls (p<0.001; Table 2). This did not

**Table 2. Vitamin D supplementation behaviour and serum 25-(OH)D levels in the biological sampling group.**

|  | Supplementing behaviour | | | | Serum 25(OH)D levels, median nmol/L (IQR); n[c] | | | |
|---|---|---|---|---|---|---|---|---|
|  | Taking supplement n[a] (%) | p-value | Dose IU/day (IQR);n[b] | p-value | No supplement | p-value | Yes supplement | p-value |
| **Disease Status** | | | | | | | | |
| MS (n = 388) | 276 (72) | <0.001 | 1600 (3200);238 | <0.001 | 38 (35); 92 | 0.06 | 82 (47); 229 | <0.001 |
| Control (n = 305) | 79 (26) | | 600 (800);63 | | 44 (21); 194 | | 68 (34); 67 | |
| **MS split by sex** | | | | | | | | |
| female | 209 (72) | 0.67 | 2000 (3084);181 | 0.52 | 38 (36); 70 | 0.51 | 82 (46); 168 | 0.65 |
| male | 67 (70) | | 1000 (4200);57 | | 38 (33); 22 | | 82 (49); 61 | |
| **MS type** | | | | | | | | |
| RRMS | 99 (73) | 0.91 | 2000 (4000);86 | 0.11 | 46 (28); 31 | 0.10 | 81 (48); 83 | 0.11 |
| SPMS | 88 (73) | | 1000 (3200);74 | | 32 (27); 26 | | 79 (48); 69 | |
| PPMS | 66 (71) | | 1428 (4000);59 | | 40 (42); 21 | | 88 (51); 58 | |
| **MS Disability** | | | | | | | | |
| low EDSS (<6) | 29 (71) | 0.78 | 2000 (4100);25 | 0.08 | 47 (32); 11 | 0.13 | 82 (68); 24 | 0.76 |
| high EDSS (≥6) | 45 (68) | | 1000 (3343);36 | | 28 (44); 16 | | 82 (63); 38 | |
| **MSIS[d]** | | | | | | | | |
| physical -Low impact | 58 (74) | 0.70 | 2857 (4000);51 | 0.04 | 46 (64); 16 | 0.69 | 88 (51); 46 | 0.36 |
| physical–High impact | 57 (77) | | 1000 (3593);44 | | 38 (50); 16 | | 80 (51); 46 | |
| psychological -Low impact | 61 (75) | 0.58 | 1800 (3750);57 | 0.77 | 46 (48); 17 | 0.76 | 83 (45); 51 | 0.66 |
| psychological-High impact | 55 (71) | | 1800 (4200);46 | | 40 (41); 17 | | 84 (48); 43 | |

[a]data was missing for the following: MS 2 participants, female MS 2 participants, MS type 2 participants, EDSS 281 participants, MSIS 236 participants;

[b]of the total n that provided supplementation data this n had a dose available;

[c]of the total n that provided supplementation data this n had serum 25(OH)D levels available;

[d]MSIS-29 scores were divided into quartiles and comparisons were made between lowest quartile (low impact) and highest quartile (high impact).

appear to be restricted to the UK MS Register population: 63% (12/19) MS participants recruited through clinics supplemented compared to just 10% (2/21) of their matched controls. There was no difference in reported rates of vitamin D supplementation across gender, MS type, disability level or score on MSIS (Table 2). Where dose data were available, MS participants (n = 238) reported a higher median vitamin D supplement dose than controls (n = 63) (1600 vs. 600 IU/day; p<0.001) (Fig 2a).

## Vitamin D supplementation in MS

Exploratory analysis of all MS questionnaire data (nMS = 1768) demonstrated that both participants with RRMS and PPMS reported taking a higher vitamin D supplement dose than participants with SPMS (median dose 2000 IU/day for both RRMS and PPMS vs 1600 IU/day for SPMS; p = 0.007) (S2 Table). Linear regression demonstrated that vitamin D supplement dose decreased with increasing years since diagnosis, although age did not appreciably affect dose (S3 Table).

(a)
(b)

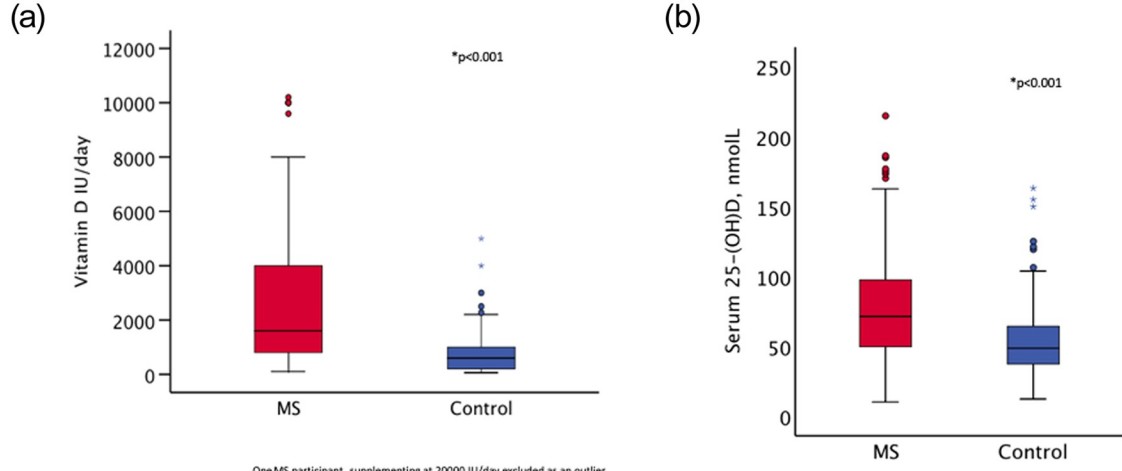

(c)

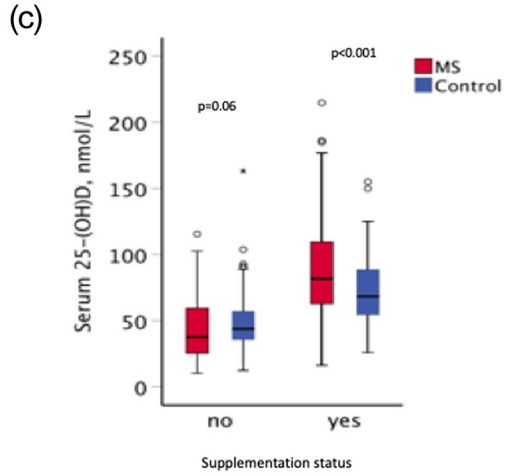

**Fig 2. Vitamin D supplementation dose and serum 25(OH)D levels in MS and control participants.** (a) Distribution of dose (IU/day) of vitamin D amongst those MS (n = 238) and control (n = 63) that take supplements. (b) Serum 25(OH)D levels (nmol/L) of MS (n = 321) and control (n = 261) participants. (c) Serum 25-(OH)D levels of MS and control split by Vitamin D supplementation status.

## Lifestyle factors influencing serum vitamin D levels

More MS participants identified as either vegetarian or vegan (11% vs 4% controls), p = 0.003. There was no difference in oily fish consumption (S4 Table). MS participants were more likely to report rarely spending time on outdoor activities (44% vs 14% controls), p<0.001 (S4 Table), which was strongly associated with disability levels. 71% (47/66) of participants with high EDSS (≥6) rarely participated in outdoor activities compared to 17% (7/41) of participants with low EDSS (<6) (p<0.001). MS participants were less likely than controls to wear sunblock (31 vs 13% "never" wear sunblock, p<0.001) (S4 Table). Females, both cases and controls, were more likely to wear cosmetic sunblock than males (24% females vs 2% males reported wearing it weekly or more, p<0.001) (S4 Table). There was no significant difference between MS and control females with respect to cosmetic sunblock usage, p = 0.09 (S4 Table).

## Serum 25(OH)D levels

Median serum 25(OH)D levels were higher in MS participants than controls: 71 vs 49nmol/L, p<0.001 (Fig 2b). MS participants were more likely to have adequate serum levels (defined as >50nmol/L) (75% MS vs 47% controls) (Table 3). There were no differences in serum vitamin D levels by gender, MS type or disability level. Subgroup analyses stratified by supplementing status demonstrated that MS participants who did not supplement (n = 92) had lower median serum 25(OH)D levels compared to non-supplementing controls (n = 194) (38 vs 44nmol/L, p = 0.06). Conversely, supplementing MS participants had higher 25(OH)D levels than supplementing controls (82 vs 68nmol/L, p<0.001) (Table 2; Fig 2c).

Solar contribution to serum 25(OH)D levels was studied using a linear regression model, which confirmed the assumption that, in the entire non-supplementing population (i.e. MS and control), latitude and time spent outdoors were significant contributors to serum 25(OH)D level ($R^2$ = 0.22, p<0.001). There was a negative association between latitude and serum 25(OH)D and positive association with time spent outdoors. Season of sampling and use of sunblock did not affect serum levels (S5 Table).

In the non-supplementing MS population increasing age had a negative association with serum 25(OH)D in a multivariable model. BMI was not associated with serum 25(OH)D levels. In the supplementing MS population there was a positive association between increasing vitamin D dose and serum 25(OH)D levels, but age, BMI or solar contributions were not associated with serum levels (S5 Table).

## Vitamin D levels in UK Biobank

People with MS in UKBB had lower median serum 25(OH)D levels than matched controls (44 vs 47 nmol/L, p<0.001). There was no difference between supplementing participants with MS vs supplementing controls (median serum 25(OH)D level 57 vs 58nmol/L). Non-supplementing people with MS had lower median serum 25(OH)D levels than either group (42nmol/L) (S6 Table). A lower proportion of people with MS took vitamin D supplementation at

**Table 3. Vitamin D status of MS and matched controls based on 25(OH)D levels.**

| Serum 25(OH)D nmol/L | Interpretation | MS (n = 322) n (%) | Control (n = 264) n (%) | p-value |
|---|---|---|---|---|
| <15 | Severe deficiency | 4 (1) | 2 (1) | p<0.001 |
| 15–30 | Deficiency | 29 (9) | 30 (11) | |
| 30.1–50 | Insufficiency | 48 (15) | 108 (41) | |
| >50 | Adequate | 241 (75) | 124 (47) | |

UKBB enrolment than in our current study, however they were still more likely to do so than the matched controls (14% vs 6%, p<0.001) (S6 Table).

## Discussion

In this case-control study we found a striking difference in vitamin D supplementation between people with MS and controls. 72% participants with MS report taking vitamin D supplements compared to just 26% of controls. Not only were MS participants more likely to take vitamin D supplements, but they also took them at higher doses, such that people with MS in the UK now have overall higher serum 25(OH)D levels than controls. When stratified by supplementation habits we found that non-supplementing people with MS had lower levels of serum 25(OH)D. These findings carry implications for any future vitamin D supplementation trial—double blind, placebo-controlled supplementation trials need to take current behavioural patterns into account, and a "treat to target" trial utilising remote sampling is likely the most feasible study design for any large-scale study.

This study is novel in its use of remote sampling technology. Given the current COVID-19 pandemic, the use of remote technologies to enable clinical trials to continue is highly relevant; we demonstrate that this is feasible in MS. The wide coverage we were able to achieve using remote sampling is particularly important when studying an environmentally sensitive endpoint such as serum 25(OH)D. The recruitment of a large pool of participants allowed us to stratify and select participants for biological sampling which represented all stages of MS with a range of disability. The use of straightforward sampling techniques carried out by participants at home allowed us to enrol all members of the MS community regardless of care centre, location or disability level.

The fact that 1722 people from a total UK MS Register population of 14,991 responded to the initial invitation to participate is worthy of note. This attrition rate is not entirely unexpected. Whilst the UK MS Register has a high number of registrants, only a proportion of these individuals return to complete surveys–around 4,000 in previous studies [22]. The relatively low rate of responses to the initial questionnaire likely reflects that this was the first UK MS Register-hosted study where participants were asked to de-anonymise themselves for research purposes, and where biological sampling was required. The rate of return of usable sample packs (58%) is in keeping with other studies requiring sample return.

This study is not without limitations. As recruitment primarily took place through a voluntary MS Register, it could be argued that this high rate of supplementation resulted from a participation/recruitment bias with an *a priori* interested population. This could have led to the recruitment of a population who were far more likely to be supplementing than the general MS population, limiting the generalisability of these results. A recent clinical trial of vitamin D supplementation in MS reported that around 17% participants were taking vitamin D supplements at baseline [9], which is closer to the proportion of supplementing participants seen in UK Biobank. However, this trial could conceivably have been enriched for patients not taking supplements due to recruitment bias. Furthermore, it could be argued that the participants in our study represent a subset of patients particularly interested in diet and lifestyle factors, and that lifestyle-aware people with MS may also be likely to have lifestyle-aware friends.

People were aware from the information sheet that the purpose of the study was to establish vitamin D levels across the UK MS population. However, no overt reference was made to either an underlying hypothesis linking vitamin D deficiency to MS or recommended intakes. The population taking part in the UK MS Register represent a more engaged and educated group with respect to vitamin D supplementation and MS. However, the recruitment of a subset of individuals directly from MS clinics across the UK enabled us to estimate bias related to

method of recruitment. Similarly high rates of supplementation were found in MS participants recruited via both means. It seems possible that these biases would also impact on recruitment to any potential vitamin D trials, and must be taken into account in any trial design.

We were unable to include measures of disease activity, such as relapse and/or MRI data in this study. EDSS data were only available on a proportion of participants. Whilst asking individuals about retrospective relapse data would have been possible, the potential for recall bias along with the influence of relapses on behaviour would have significantly limited the interpretation of data collected in this way. The finding that supplementation rates decreased with increasing years since MS diagnosis is interesting; potential reasons for this could include either individuals stopping supplementation as their MS progresses, or alternatively it could reflect changing advice from neurologists. This finding may be worthy of additional studies examining the drivers of both starting and of stopping supplementation.

Participant recruitment of age and sex-matched controls may have induced bias related to overmatching, however the exclusion of household controls mitigates this to some degree. Whilst similarities may remain around socioeconomic status and other lifestyle factors, we see that the impact of differential vitamin D status far outweighs this.

Whilst the UKBB population demonstrated a lower rate of vitamin D supplementation amongst people with MS compared to our current study, vitamin D supplementation was still significantly higher than in controls. The reason(s) for the discrepancy between vitamin D usage between the current study and the UK Biobank population is unclear, but at least some of this difference may be attributed to the changes in vitamin D usage over the last 10 years. UKBB baseline data was collected 10–14 years ago, and attitudes towards vitamin D supplementation in the UK have changed significantly over this time [23]. Our clinical experience reflects this finding that overall supplementation has increased over the past 10–15 years, and advice regarding vitamin D has now become mainstream, particularly in MS clinics. The reasons for the difference between supplementation in the UK MS Register and in UK Biobank is therefore likely multifactorial.

Due to the cross-sectional nature of this observational case-control study we are unable to make inferences with regard to vitamin D status and disease progression. However, the potential to re-recruit via the same online platform for follow-up remains. The use of self-reported behaviours is a further limitation, however, the dose-response to vitamin D supplementation and validation using photographs of supplements overcome this to some degree. The UK MS Register population is predominantly White British [17] and this study needs to be replicated in an ethnically diverse population. Finally, whilst the return rate of biological samples was high for a survey-based study, it remains significantly lower than in direct sampling studies, and this must be considered in future remote sampling studies.

In conclusion we have characterised the behaviours influencing vitamin D and carried out a detailed analysis of the vitamin D status across the UK MS population. People with MS are more likely to supplement with vitamin D and at higher doses than matched controls. After supplementation behaviours, outdoor activity had the most significant impact on serum 25 (OH)D levels. The solar contribution to vitamin D levels was evidenced through both positive association with time spent outdoors and a negative association with increasing latitude. This study underlines the importance of considering participant lifestyle, behavioural and baseline vitamin D status when considering the design of interventional trials using vitamin D in MS.

## Supporting information

**S1 Fig. Distribution of study participants across the UK.** (a) The distribution of the 1768 study participants who provided questionnaire data. (b) The distribution of the MS

participants selected to receive biological sampling kits.
(DOCX)

**S1 Table. Vitamin D supplementation behaviour and serum 25(OH)D levels in MS cases.**
(DOCX)

**S2 Table. Multivariable analysis of factors influencing vitamin D dose of participants.**
(DOCX)

**S3 Table. Lifestyle factors and behaviours known to influence serum vitamin D in those who provided biological samples.**
(DOCX)

**S4 Table. Multivariable analysis of variables influencing vitamin D serum 25(OH)D levels.**
(DOCX)

**S5 Table. Demographic details of those included in the UK Biobank study.**
(DOCX)

**S6 Table. Validation cohort taken from UK Biobank.**
(DOCX)

**S1 Checklist. STROBE statement—Checklist of items that should be included in reports of case-control studies.**
(DOC)

## Author Contributions

**Conceptualization:** Gavin Giovannoni, Ruth Dobson.

**Data curation:** Rod Middleton.

**Formal analysis:** Mark Jitlal, Benjamin Meir Jacobs.

**Funding acquisition:** Ruth Dobson.

**Investigation:** Nicola Vickaryous, Siddharthan Chandran, Niall John James MacDougall, Gavin Giovannoni, Ruth Dobson.

**Methodology:** Nicola Vickaryous, Mark Jitlal, Benjamin Meir Jacobs, Siddharthan Chandran, Ruth Dobson.

**Resources:** Rod Middleton.

**Software:** Rod Middleton.

**Supervision:** Ruth Dobson.

**Writing – original draft:** Nicola Vickaryous.

**Writing – review & editing:** Mark Jitlal, Benjamin Meir Jacobs, Rod Middleton, Siddharthan Chandran, Niall John James MacDougall, Gavin Giovannoni, Ruth Dobson.

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
