## [Decision Letter · Decision Letter 0]

12 Nov 2020

PONE-D-20-32149

Remote testing of vitamin D levels across the UK MS population – a case control study

PLOS ONE

Dear Dr. Dobson,

Thank you for submitting your manuscript to PLOS ONE. After careful consideration, we feel that it has merit but does not fully meet PLOS ONE’s publication criteria as it currently stands. Therefore, we invite you to submit a revised version of the manuscript that addresses the points raised during the review process.

We look forward to receiving your revised manuscript.

Kind regards,

Sreeram V. Ramagopalan

Academic Editor

PLOS ONE

Journal Requirements:

4. We note that Supp Fig 1 in your submission contains map images which may be copyrighted. All PLOS content is published under the Creative Commons Attribution License (CC BY 4.0), which means that the manuscript, images, and Supporting Information files will be freely available online, and any third party is permitted to access, download, copy, distribute, and use these materials in any way, even commercially, with proper attribution. For these reasons, we cannot publish previously copyrighted maps or satellite images created using proprietary data, such as Google software (Google Maps, Street View, and Earth). For more information, see our copyright guidelines: http://journals.plos.org/plosone/s/licenses-and-copyright.

(1) You may seek permission from the original copyright holder of Supp Fig 1 to publish the content specifically under the CC BY 4.0 license. 

Reviewers' comments:

Reviewer's Responses to Questions

**Comments to the Author**

1. Is the manuscript technically sound, and do the data support the conclusions?

Reviewer #1: Yes

2. Has the statistical analysis been performed appropriately and rigorously? 

Reviewer #1: Yes

3. Have the authors made all data underlying the findings in their manuscript fully available?

Reviewer #1: No

4. Is the manuscript presented in an intelligible fashion and written in standard English?

Reviewer #1: Yes

5. Review Comments to the Author

Reviewer #1: The authors present data of a case-control study on determinants of vitamin D supplement use in MS. The study has methodological novelty (in the MS field) of remote sampling of vitamin D status with dried blood spots, the case-control association study on serum 25(OH)D levels and disease/ dietary status in MS has been performed multiple times previously and is confirmatory. Nevertheless, this confirmatory study has been performed well and sources of bias are discussed. There are no major novel conclusions drawn from this study, and some points require further clarification.

Major points

* A major missing disease-related endpoint is the occurrence of relapses, especially since this endpoint has been associated with vitamin D status in prior studies (Simpson et al., Ann Neurol 2010). The presence of relapses as signs of disease activity may also motivate patients to start or continue taking supplements. Do the authors have data available on recent relapses rate/ occurrence of relapses the prior years, and its influence on vitamin D supplementation?

*Please clarify how the intake of vitamin D was inquired: current intake? Intake the previous month? Did the authors sample the duration of vitamin D supplement intake? Data suggest that people with a longer disease duration may also stop somewhere during a prolonged disease course of MS.

*The issue of participation bias requires some more emphasis: 1722/14991 individuals with MS participated, sampling kits were distributed to 3 (of N=?) MS clinics. The participant may be a subset of patients most interested in diet and lifestyle factors, participating center may be more skewed towards promoting a epidemiologically-favorable (holistic) lifestyle approach, and lifestyle-aware people with MS may also be likely to have lifestyle-aware friends. This is also suggested by the higher intake of vitamin D supplements in both MS and control groups compared the UKBB/ matched control cohort.

*Of the 600 kits send out, 388 were returned. With 42% not participating, a supplementary figure disclosing the characteristics of the not-participating patients would benefit the data. Table one suggests that people with relapsing remitting MS and a relatively low EDSS score were less likely to participate in this study. The generalizability of the data to this specific group can be questioned.

*The authors only report the number of current smokers. Do the authors have also data on ‘ past smokers’, to explore whether the selected sample may be more prone to adapt lifestyle-changes when compared to the entire cohort?

Minor points

*Pg4, line 5. Please introduce the abbreviation MS.

*Pg 4, line 25. Several recent efforts to perform clinical trials on vitamin D supplements in MS have been made, including Dor et al., MSJ-ETC 2020, Hupperts et al., Neurology 2019, and Camu et al., N2 2019. These studies did not provide robust evidence either. The provided older ref [8] ignores these studies, please update.

*The sampled number of vitamin D supplements taking people may be an overestimation due to several sources of bias. The authors provide a very limited reflection of their data on for instance included clinical trial populations. Although clinical trial populations may be enriched for patients not taking supplements, a recent vitamin D supplementation study in RRMS reported 16.44-17.7% of participants to take vitamin D supplements at baseline (Hupperts et al., Neurology 2020).

6. PLOS authors have the option to publish the peer review history of their article (what does this mean?). If published, this will include your full peer review and any attached files.

Reviewer #1: No

---

## [Author Response · Author response to Decision Letter 0]

20 Nov 2020

Response to reviewer and editorial comments

We would like to thank both the editor and reviewer for the time taken to read this manuscript and provide comments. Some of the editorial comments are addressed in the cover letter, particularly those regarding data sharing, as instructed in the comments. We have also commented on these below for clarity. 

Specific comments are addressed below. 

Editorial comments: 

The manuscript has been revised in line with the style requirements

This has been addressed in the manuscript, with an expanded datasharing statement. This is also discussed in the cover letter. In brief, there are ethical restrictions on data sharing without application due to the personal and sensitive nature of some of the data that is available via the UK MS Register. 

Reviewer comments: 

Major points

* A major missing disease-related endpoint is the occurrence of relapses, especially since this endpoint has been associated with vitamin D status in prior studies (Simpson et al., Ann Neurol 2010). The presence of relapses as signs of disease activity may also motivate patients to start or continue taking supplements. Do the authors have data available on recent relapses rate/ occurrence of relapses the prior years, and its influence on vitamin D supplementation?

We agree with the referee that relapse data would have added significant depth to this study. However, unfortunately this data is not routinely prospectively collected by the UK MS Register. We were concerned that asking retrospectively about relapses and vitamin D might introduce bias on a number of levels, and provide data that were subject to misinterpretation. We therefore elected to focus on current measures and outcomes only. We have explicitly discussed this in the conclusions. 

*Please clarify how the intake of vitamin D was inquired: current intake? Intake the previous month? Did the authors sample the duration of vitamin D supplement intake? Data suggest that people with a longer disease duration may also stop somewhere during a prolonged disease course of MS.

We asked participants to provide their current vitamin D intake (dose and frequency). As the primary aim of the study was to establish current patterns and influences, we did not seek to examine behavioural change over time. Whilst we agree with the reviewer that this would be interesting, it was beyond the scope of this study. As with all retrospective work, recall bias would be a concern, along with taking into account variation in advice over both geographical area and time – advice from neurologists has shifted over time. 

Similarly, the observation that supplementation rates decreased with increasing years since MS diagnosis is interesting; potential reasons for this could include either individuals stopping supplementation as their MS progresses, or alternatively it could reflect changing advice from neurologists. As this finding was unexpected we can only speculate as to the reasons underlying this. We have highlighted this in the conclusions. 

*The issue of participation bias requires some more emphasis: 1722/14991 individuals with MS participated, sampling kits were distributed to 3 (of N=?) MS clinics. The participant may be a subset of patients most interested in diet and lifestyle factors, participating center may be more skewed towards promoting a epidemiologically-favorable (holistic) lifestyle approach, and lifestyle-aware people with MS may also be likely to have lifestyle-aware friends. This is also suggested by the higher intake of vitamin D supplements in both MS and control groups compared the UKBB/ matched control cohort.

We have now discussed this in more detail in the conclusions. Our clinical experience has been that overall supplementation has increased over the past 10-15 years, and advice regarding vitamin D has now become mainstream. The reasons for the difference between supplementation in the UK MS Register and in UK Biobank is therefore likely multifactorial – and we have been more explicit about this in the conclusions. 

*Of the 600 kits send out, 388 were returned. With 42% not participating, a supplementary figure disclosing the characteristics of the not-participating patients would benefit the data. Table one suggests that people with relapsing remitting MS and a relatively low EDSS score were less likely to participate in this study. The generalizability of the data to this specific group can be questioned. 

Table S1 now illustrates the cohorts returning samples, and those who did not. We have commented in the text on the significant differences in characteristics between these two groups – namely those not returning packs were slightly younger and more likely to be male. There was no significant difference in disability level or MS type. 

The difference in the biological sampling group represents the stratified random sampling rather than response rates. As part of this approach, we attempted to ensure that we had a balance across MS disease types and EDSS in order that we could make meaningful comparisons across pre-specified subgroups. If we had sought to represent the initial respondents we would not have had enough individuals in some cells in order to be able to perform such analyses. This has now been made clearer in the methods 

*The authors only report the number of current smokers. Do the authors have also data on ‘ past smokers’, to explore whether the selected sample may be more prone to adapt lifestyle-changes when compared to the entire cohort?

Unfortunately, these data were not available, although we agree that this analysis would have been interesting. 

Minor points

*Pg4, line 5. Please introduce the abbreviation MS.

This change has been made

*Pg 4, line 25. Several recent efforts to perform clinical trials on vitamin D supplements in MS have been made, including Dor et al., MSJ-ETC 2020, Hupperts et al., Neurology 2019, and Camu et al., N2 2019. These studies did not provide robust evidence either. The provided older ref [8] ignores these studies, please update.

These references have been added. 

*The sampled number of vitamin D supplements taking people may be an overestimation due to several sources of bias. The authors provide a very limited reflection of their data on for instance included clinical trial populations. Although clinical trial populations may be enriched for patients not taking supplements, a recent vitamin D supplementation study in RRMS reported 16.44-17.7% of participants to take vitamin D supplements at baseline (Hupperts et al., Neurology 2020).

We have tried to expand the conclusions to lay out the sources of bias more clearly. We have included the reference suggested, and highlighted that the proportion of people with MS who are supplementing in this study reflects the proportion supplementing at UK Biobank baseline.

---

## [Editor Report · Decision Letter 1]

16 Dec 2020

Remote testing of vitamin D levels across the UK MS population – a case control study

PONE-D-20-32149R1

Dear Dr. Dobson,

We’re pleased to inform you that your manuscript has been judged scientifically suitable for publication and will be formally accepted for publication once it meets all outstanding technical requirements.

Kind regards,

Sreeram V. Ramagopalan

Academic Editor

PLOS ONE
---

## [Editor Report · Acceptance letter]

18 Dec 2020

PONE-D-20-32149R1 

Remote testing of vitamin D levels across the UK MS population – a case control study 

Dear Dr. Dobson:

I'm pleased to inform you that your manuscript has been deemed suitable for publication in PLOS ONE. Congratulations! Your manuscript is now with our production department. 

Kind regards, 

on behalf of

Dr. Sreeram V. Ramagopalan 

Academic Editor

PLOS ONE